# The Normobaric Oxygen Paradox—Hyperoxic Hypoxic Paradox: A Novel Expedient Strategy in Hematopoiesis Clinical Issues

**DOI:** 10.3390/ijms24010082

**Published:** 2022-12-21

**Authors:** Michele Salvagno, Giacomo Coppalini, Fabio Silvio Taccone, Giacomo Strapazzon, Simona Mrakic-Sposta, Monica Rocco, Maher Khalife, Costantino Balestra

**Affiliations:** 1Department of Intensive Care, Hôpital Universitaire de Bruxelles (HUB), 1070 Brussels, Belgium; 2Institute of Mountain Emergency Medicine, Eurac Research, 39100 Bolzano, Italy; 3Institute of Clinical Physiology—National Research Council (CNR-IFC), 20162 Milano, Italy; 4Dipartimento di Scienze Medico Chirurgiche e Medicina Traslazionale, Sapienza University of Rome, 00189 Rome, Italy; 5Department of Anesthesiology, Institut Jules Bordet, Université Libre de Bruxelles (ULB), 1070 Brussels, Belgium; 6Environmental, Occupational, Aging (Integrative) Physiology Laboratory, Haute Ecole Bruxelles-Brabant (HE2B), 1050 Brussels, Belgium; 7Anatomical Research and Clinical Studies, Vrije Universiteit Brussels (VUB), 1090 Brussels, Belgium; 8DAN Europe Research Division (Roseto-Brussels), 1020 Brussels, Belgium; 9Physical Activity Teaching Unit, Motor Sciences Department, Université Libre de Bruxelles (ULB), 1050 Brussels, Belgium

**Keywords:** hypoxia, hyperoxia, HIF-1α, oxygen biology, human, stimulus, cancer, intensive care, rehabilitation, human performance, preconditioning, pre-habilitation

## Abstract

Hypoxia, even at non-lethal levels, is one of the most stressful events for all aerobic organisms as it significantly affects a wide spectrum of physiological functions and energy production. Aerobic organisms activate countless molecular responses directed to respond at cellular, tissue, organ, and whole-body levels to cope with oxygen shortage allowing survival, including enhanced neo-angiogenesis and systemic oxygen delivery. The benefits of hypoxia may be evoked without its detrimental consequences by exploiting the so-called normobaric oxygen paradox. The intermittent shift between hyperoxic-normoxic exposure, in addition to being safe and feasible, has been shown to enhance erythropoietin production and raise hemoglobin levels with numerous different potential applications in many fields of therapy as a new strategy for surgical preconditioning aimed at frail patients and prevention of postoperative anemia. This narrative review summarizes the physiological processes behind the proposed normobaric oxygen paradox, focusing on the latest scientific evidence and the potential applications for this strategy. Future possibilities for hyperoxic-normoxic exposure therapy include implementation as a synergistic strategy to improve a patient’s pre-surgical condition, a stimulating treatment in critically ill patients, preconditioning of athletes during physical preparation, and, in combination with surgery and conventional chemotherapy, to improve patients’ outcomes and quality of life.

## 1. Introduction

Oxygen is essential to support cellular biology and most vital cell reactions. Hypoxemia is a condition where below-normal levels of oxygen are dissolved in the blood and may result in poor oxygen delivery to peripheral tissues and organs, causing a supply/demand discrepancy called hypoxia [1]. If this state persists, tissues may develop hypoxemic stress, leading to organ dysfunction and permanent functional impairment [2]. To prevent this kind of damage, hypoxia is a key inducer of cellular gene expression, promoting many processes (aimed at improving oxygen delivery), such as angiogenesis, stem cell proliferation and differentiation, but also cellular protection and repair, or cell death. Most of this gene expression is guided by the activity of transcription factors called Hypoxia Inducible Factors (HIF) [3].

To prevent or counteract hypoxia, oxygen, which is considered as inexpensive and safe, is one of the most widely used treatments in clinical settings, especially in the intensive care unit (ICU) [4]. However, hyperoxemia is not constantly assessed. In fact, hypoxemia is usually monitored using peripheral oxygen saturation (SpO_2_), which cannot detect whether the oxygen level is too high. Oxygen administration is therefore not often titrated to achieve normoxia. Consequently, many patients in the ICU may be exposed to episodes of hyperoxemia (high PaO_2_), generally considered more acceptable by clinicians than hypoxia. 

Even though hyperoxia may seem harmless, it can have detrimental effects even at modest levels if prolonged, especially in critically ill patients. In fact, hyperoxia is associated with increased mortality [5], and other possible adverse effects, such as reduced mucociliary clearance and atelectasis [6], pulmonary vascular vasoconstriction [7], which may further limit oxygen delivery and cause direct tissue damage [8], and neurotoxicity [9]. Nevertheless, administration of high partial pressure of oxygen is sometimes necessary for specific treatments such as hyperbaric oxygen therapy [10] and pre-oxygenation maneuvers to prevent hypoxia (during procedures such as bronchoscopy or oro/nasotracheal intubation where brief moments of apnea cannot be avoided) [11].

There is currently a move toward not only targeting normoxia but using intermittent hyperoxic stimuli with cyclical exposure to hyperoxygenation [12]. An intermittent hyperoxic stimulus is defined as an elevated oxygen concentration supplied for a limited period, followed by a return to a lower oxygen concentration, repeated for several days or even several times per day [12,13]. The decrease from a hyperoxic level has been shown to act paradoxically because the cells respond to it as they would respond to a hypoxic state [14,15]. It appears that fluctuations in oxygen concentration are translated by cells as a lack of oxygen and can trigger the hypoxic stress response even when there is no persistent hypoxia. This phenomenon happens especially with small and cyclical variations in partial oxygen pressure [16,17,18] and is linked with increased production of erythropoietin (EPO) in humans [15,19].

This review will discuss the main studies related to this “normobaric oxygen paradox” and its mechanism with a specific concern for clinical issues in hematopoietic physiology.

## 2. Hypoxia

### 2.1. Biological Cellular Response: Hypoxia Inducible Factors (HIFs)

Hypoxia generates a natural and multi-aspect response inside the organism through the involvement of various systems (e.g., hematopoietic, metabolic, respiratory, cardiovascular) and biological processes with specific cellular reactions and temporal regulation, all aimed at maintaining adequate tissue oxygenation [20,21].

The hypoxic stress inside cells activates a family of transcriptional factors called Hypoxia Inducible Factors (HIF-1α, HIF-2α, HIF-3α) [3,22]. First discovered at the Johns Hopkins University by Semenza, Kaelin and Ratcliffe more than 30 years ago [23], Semenza and colleagues isolated and purified HIF-1, confirming the presence of two subunits: HIF-1α and HIF-1β [24,25]. Later, it was shown that there is a family of HIFs: the subunit α could be one among HIF-1α, HIF-2α and HIF-3α; the β subunit is represented by one protein (HIF-1β). HIF-1α is widely expressed in all body tissues, while HIF-2α and HIF-3α are only detected in a few specific tissues [3,26,27]. HIFs modulate the response to hypoxia by prompting the expression of hundreds of genes that are involved in metabolism regulation, angiogenesis, cell growth/death, cell proliferation and division, glycolysis, microbial infection, tumor genesis and metastasis, oxygen consumption, erythrocyte production, mitochondrial metabolism, immune and inflammatory response [28,29,30]. HIFs can be considered one of the main regulators of O_2_ detection, driving cellular adaptation to a specific oxygen level. 

### 2.2. HIF Regulation

In the presence of oxygen, prolyl hydroxylase domain (PHD) proteins which contain oxygen-sensing hydroxylases, continuously hydroxylase specific residuals on the α subunit of HIF (Figure 1) [31]. 

When HIF-1α is hydroxylated, it becomes a viable target for von Hippel–Lindau (VHL) protein, which activates its ubiquitin ligase system, leading to the proteasomal degradation of HIF-α [32,33]. Moreover, VHL acts as a repressor of HIF-1α by binding the inhibitory domain, as does FIH-1 (Factor Inhibiting HIF-1), which interacts with both HIF-1α and VHL via independent binding sites [34]. 

By contrast, in hypoxic conditions, HIF-1α is stabilized, driving its translocation to the nucleus, its dimerization with HIF-1β, and its binding to hypoxia response elements (HREs) sequence on DNA, thus regulating the transcription of around 400 genes, including the gene for EPO [29,35,36].

As a counterpart of HIF transcriptional activity, the nuclear factor erythroid 2-related factor 2 (NRF2) has a central role, coordinating the activation of a vast array of cytoprotective genes. In response to different activating stimuli following disturbances of the cellular redox status, NRF2 is stabilized, and it translocates to the nucleus where it binds to antioxidant response elements (AREs) in the promoter regions of genes encoding for antioxidant, cytoprotective proteins including glutathione synthesis, as well as glutathione reductase and enzymes involved in NADPH regeneration, xenobiotic detoxification and heme metabolism [37,38].

Despite the importance of these other actors in the cellular redox status control, this review will focus on HIF activity redirecting the readers to more specialized publications for more in-depth analysis.

### 2.3. Reactive Oxygen Species (ROS)

In this complex mechanism, mitochondria have a central role, representing the core of cellular respiration. Here, oxygen reacts with glucose to produce ATP (adenosine triphosphate), which serves as energy for the body [39]. In particular, oxygen molecules sit at the end of the electron transport chain, necessary for ATP production, as electron acceptors. During this process, a minimum amount of reactive oxygen species (ROS) is produced. These represent a group of highly reactive molecules characterized by unpaired electrons derived from oxygen reduction, which can quickly react with other cellular molecules [40,41]. 

The detrimental effects of the reactive species of oxygen are counteracted through the activation of NRF2, which tightly regulates an antioxidant response through the encoding of several genes, as forementioned. Thus, ROS are kept under control by scavengers, such as superoxide dismutases (SODs, a group of metalloproteins, which catalyzes the reduction of superoxide anions to hydrogen peroxide) and glutathione (GSH, a substrate of glutathione reductase that reduces hydroperoxides to the corresponding alcohol, oxidizing itself to the oxidized disulfide form, GSSG) [42,43,44]. 

Nowadays, ROS are no longer considered just as harmful, rather are they are believed to play a role as mediators in physiological signaling, linked to the regulation of the HIF system [45,46]. HIF, NRF2, ROS, scavengers, and other regulators appear to be tightly linked, highlighting the importance of cellular control of oxygen levels, which still needs to be fully understood.

Not every molecule and biological reaction involved in the fascinating process of cellular response to hypoxia have been cited here (there are indeed many other participants including peroxiredoxin reductase/oxygenases and catalase, to name just a few), both for the sake of simplicity and because out of the focus of our review which is not to provide a full biological molecular description (thoroughly described elsewhere [38,47,48]) but to summarize new experimental evidence of practical application of the normobaric oxygen paradox and illustrate the perspective of its desirable future clinical applications.

Hypoxic Inducible Factor 1 (HIF-1) is a heterodimer composed of 2 subunits: HIF-1α in the cytosol (HIF-1α is widely expressed in all body tissues, whereas HIF-2α and HIF-3α are only detected in a few specific tissues), and HIF-β in the nucleus. During normoxia prolyl hydroxylase domain (PHD) proteins which contain oxygen-sensing hydroxylases, continuously hydroxylase specific residues on the α subunit of HIF (blue circles). When HIF-1α is hydroxylated, it becomes a viable target for von Hippel–Lindau (VHL) protein, which activates its ubiquitin ligase system, leading to the proteasomal degradation of HIF-1α. Moreover, VHL acts as a direct repressor of HIF-1α by binding the inhibitory domain, as does FIH-1 (factor inhibiting HIF-1). Reactive species of oxygen (ROS) are kept under control by scavengers such as superoxide dismutases (SODs) and glutathione. The latter acts as a scavenger of H_2_O_2_, oxidizing from GSH to GSSG thanks to the activity of glutathione peroxidase. The reduction in GSH is led by glutathione reductase, which consumes NADPH. Moreover, in response to different activating stimuli following disturbances of the cellular redox status, NRF2 is stabilized, and it translocates to the nucleus where it binds to antioxidant response elements in the promoter regions of genes encoding for antioxidant, cytoprotective proteins including glutathione synthesis and reduction, as well as enzymes involved in NADPH regeneration.

## 3. The Normobaric Oxygen Paradox

### 3.1. Background 

It has been proposed that relative changes in oxygen availability, rather than an absolute hypoxic or hyperoxic value, play an essential role in the transcriptional effects of HIF [49]. The *normobaric oxygen paradox* (NOP, also called *hyperoxic-hypoxic paradox* when considered in other than normobaric pressure) postulates that a period of hyperoxemia (obtained with normobaric or hyperbaric oxygen inhalation) followed by a return to normoxia would be interpreted by our cells as an oxygen shortage, thus potentially triggering a HIF-1α regulated gene synthesis cascade, including synthesis of EPO. 

This paradox was proposed for the first time in 2006 by Balestra et al. [15]. In that study, the authors found that intermittent hyperoxia/normoxia exposure induced EPO synthesis. The authors reported that hyperbaric oxygen (HBO) was an essential depressor of serum EPO levels up to 24 h after hyperbaric treatment. In 2012, Cimino and colleagues reported that reducing the O_2_ concentration from hyperoxic to normoxic levels could stimulate HIF-1α expression in human umbilical endothelial cells [14]. They also showed an increase in hemoglobin concentration in healthy volunteers (*n* = 24) when exposed to 30 min cycles of hypoxia (FiO_2_ 0.15) and hyperoxia (FiO_2_ 1.0) every other day for 10 days (5 sessions). These studies confirmed the hypothesis that a hyperoxic stimulus may re-create the benefits of a hypoxic-like response (through the production or non-inhibition of HIF-1α) without inducing potentially harmful hypoxic status. 

### 3.2. The Proposed Mechanism 

In a recent review, Hadanny and Efrati [33] redescribed the same explanation for the normobaric oxygen paradox but focused on hyperbaric oxygen exposure. However, intermittent Hyperbaric or Normobaric oxygen exposures will elicit the same reactions. The general process proposed (Figure 2) lies in the fundamental cellular mechanism of adaptation to hypoxia, namely on the O_2_ availability of free radicals. The ratio of ROS to scavenging capacity is the key to understanding the process. The different steps of this phenomenon can be summarized as follows:

(1)After exposure to hyperoxia, the increased presence of ROS causes an increase in activity of the glutathione synthetase enzyme. This increase in scavenger molecules keeps the oxidative conditions of the cells under control, thus preventing the potential harm caused by reactive species to DNA and other pivotal cellular processes.(2)When returning to normoxia, normalization of oxygen levels and therefore of ROS is rapidly established, but activity of the scavenger power of the cells remains high for a longer period, exceeding the amount of ROS normally produced in the presence of a physiological concentration of oxygen. When the hyperoxic stimulus is interrupted, the more significant scavenger presence than ROS could drive a hypoxia-like cellular response as lower reactive oxygen species molecules are available. Therefore, less HIF undergoes proteasomal degradation, promoting the transcription of EPO, vascular endothelial growth factor (VEGF), and all the other genes linked to the HIF cascade.(3)Cyclical hyperoxic exposure causes a decrease in the ROS/scavenger ratio until this gradually becomes similar to the balance present under hypoxic conditions. From a molecular point of view, a reduction of hyperoxia generates a hypoxia-mimicking state by decreasing the percentage of ROS/scavenging capacity.

However, several remarks may be raised against this mechanism. 

First, several studies evaluated the possible effects of ROS on the complex regulation of HIF and identified conflicting roles. Some results support the possible HIF stabilization through the inhibition of the proteins which are supposed to degrade it; others indicate that ROS induce the degradation of HIF through the proteasome pathway [50,51,52]. Moreover, the cellular response to hyperoxia-normoxia is mastered by the interplay between the activation of two transcription factors, HIF and NRF2. These, some inflammatory reactions, and other factors, such as NF-κB, are of course of major interest in the mechanism responding to hyperoxia. Their dynamic balance could lead and potentiate the whole mechanism [16,17].

Surely, the biological mechanism which lies beneath this phenomenon and all of the delicate regulations of the cellular response to oxygen and oxidative stress is complicated and has not been completely clarified yet. The significance of ROS in mediating this response remains unclear. Many other actors have a role in the redox balance of the cell and may also play a role in the explanation of the normobaric oxygen paradox.

The upper third of Figure 2 describes what happens during normoxia (redrawing of Figure 1). After a hyperoxic exposure (middle third), the increased presence of reactive oxygen species (ROS) determines an increase in the scavenging activity, which keeps the oxidative conditions of the cells under control, thus preventing the potential harm caused by reactive species to DNA and other pivotal molecules. After returning to normoxia from a hyperoxic state (lower third), extra scavengers induced by hyperoxia neutralize all the reactive oxygen species (ROS). The more significant scavenger presence than ROS could drive a hypoxia-like cellular response: less HIF-1α is hydroxylated, and thus it can now dimerize and translate for several genes. The duration of the hyperoxic stimulus, its frequency and the exact timing, which optimally elicits this mechanism, are still unknown (represented by the clock and the question mark). 

## 4. Methods 

A narrative approach was chosen for this review. A literature search was initially performed in PubMed, Scopus, and Google Scholar to identify studies, conducted in the last 20 years, that explored the normobaric oxygen paradox. The following search string was used: (“Hyperoxic” AND “Hypoxic”) OR (“Normobaric” AND (“oxygen” OR “hyperoxic”)) AND “paradox”. The review was focused on humans both experimental and in-hospital settings with a preference for a clinical approach. We considered studies testing hyperbaric or normobaric hyperoxic–normoxic stimulus in adult healthy subjects and studies which evaluated potential oxygen effects on adult patients scheduled for general or cardiac surgery. Although it was of interest to our group, only a few studies involving critically ill patients have been found. The search was restricted to articles published in English in peer-reviewed journals. No restriction on study design was imposed. Abstract presentations, conference proceedings, and reviews were excluded.

Studies were manually selected based on title and abstract. Selected studies were read thoroughly to identify those suitable for inclusion in this narrative review. We extracted the demographic and experimental data from the selected studies. For each study, the following relevant information was extracted and summarized: characteristics of the investigated population; oxygen administration protocols (hyperbaric vs. normobaric; hyperoxia to normoxia or mild hypoxia); the experimental and/or clinical settings of application; and the main results of the studies in terms of body response to hypoxia and enhancing effect on HIF- 1α pathway.

A schematic flow of the study selection is represented in Figure 3.

## 5. Human Studies

Available studies dealing with NOP administration to humans are shown in Table 1.

In 2006, 16 healthy volunteers were studied by Balestra et al. before and after a 2 h period of breathing 100% normobaric oxygen and a 90-min period of breathing 100% hyperbaric oxygen at 2.5 ATA [15]. Serum EPO concentrations were measured at various time points during the subsequent 24–36 h. The authors observed a 60% increase in serum EPO 36 h after normobaric oxygen. By contrast, a 53% decrease in serum EPO concentration was observed 24 h after hyperbaric oxygen, suggesting that normobaric oxygen evokes a higher response in EPO production than hyperbaric oxygen. These results were unexpected since one could imagine that a higher stimulus should induce a higher response, paradoxically it was not the case. For this reason, they introduced the term “*normobaric oxygen paradox*”.

In 2011, Keramidas et al. published conflicting results [53]. In this single-blinded crossover trial, 10 healthy male volunteers breathed for 2 h ambient air (NOR group) first and then 100% normobaric O_2_ (HYPER group). Blood samples were collected pre, mid, and post-exposure, and at 3, 5, 8, 24, 32, 48, 72, 96 h, 1 and 2 weeks after the exposure to determine serum EPO concentration. The authors observed an increase in serum EPO concentration at 8 and 32 h after ambient air (by 58% and 52%, respectively, *p* < 0.05), but in the discussion, this increase was attributed to natural EPO diurnal variation. Conversely, in the HYPER group, there was a 36% decrease in EPO 3 h after the exposure (*p* < 0.05). Moreover, EPO concentration was significantly lower in the HYPER than in the NOR group at 3, 5 and 8 h after the breathing intervention. Despite these significant results, it must be noted that the authors did not adjust their results to individual diurnal variations in EPO as carried out by others [15].

Several studies have been performed in the clinical setting. Lafère and colleagues [56] found that the normobaric oxygen paradox effectively increased the reticulocyte count after traumatic hip surgery. Patients were randomly assigned to a control group (*n* = 40) receiving 30 min of air or an O_2_ group (*n* = 40) where patients were exposed to 100% normobaric oxygen 15 L/min for 30 min every day from the first postoperative day until discharge. On day 7, the O_2_ group showed a significant increase in reticulocyte count and percent variation (184.9% ± 41.4%) compared to the air group (104.7% ± 32.6%). Even though no differences were found in hemoglobin or hematocrit levels, red blood cell (RBC) transfusions were significantly lower in the O_2_ group than in the air group. However, this latter finding was not attributed to the increase in reticulocytes, since blood was administered before that variation, but probably to a better anemia/hypoxia tolerance or better coagulation related to oxygen stimuli.

Ciccarella et al. [54] presented, in a letter to the editor, results from a prospective, randomized, double-blind pilot trial, in which they evaluated 20 cardiac surgery patients divided into two groups. In the first group (*n* = 10), patients received FiO_2_ 1.0 of normobaric O_2_ for 2 h, followed by exposure to FiO_2_ 0.5 normobaric O_2_ post-cardiac surgery. The second group (*n* = 10) received FiO_2_ 0.5 of normobaric O_2_ for 2 h. The slope of the increase in the plasmatic EPO level was significantly higher in the FiO_2_ 1.0 group than in the FiO_2_ 0.5 group, confirming that this stimulus may be helpful to enhance the endogenous production of EPO (eliciting cardioprotection and neuroprotection) in postoperative conditions.

In 2017, Donati et al. evaluated 20 hemodynamically stable, mechanically ventilated patients with inspired oxygen concentration (FiO_2_) ≤ 0.5 and PaO_2_/FiO_2_ ≥ 200 mmHg; the patients had a 2-h exposure to hyperoxia (FiO_2_ 1.0) [19]. A group of 20 patients with similar characteristics was chosen as a control. Blood samples were collected from both groups at 24 and 48 h to measure serum EPO concentration, and at baseline (t0), after 2 h of hyperoxia (t1) and 2 h after return to the initial FiO_2_ concentration (t2) to measure serum glutathione and ROS levels. In addition, the microvascular sublingual response to hyperoxia was assessed. Serum ROS increased transiently at t1, and glutathione increased at t2. Interestingly, EPO levels increased in the hyperoxia group (*p* < 0.05) and were significantly higher at 48 h compared to baseline; no changes were seen in the control group. By contrast, there was no increase in the reticulocyte count, which decreased after 48 h, apparently in conflict with the results of Lafère et al. [56]. However, as discussed by the authors themselves, the 48-h observational period is too short to detect an increase in reticulocyte count and Hb, after one single hyperoxia exposure.

In 2021, Fratantonio et al. [17] randomized 12 healthy adult individuals to three groups of different oxygen FiO_2_ exposure: the first group received one hour of FiO_2_ 0.3 (mild hyperoxia, MH), the second received one hour of FiO_2_ 1.0 for normobaric hyperoxia (high hyperoxia, HH), and the third received one hour of FiO_2_ 1.4 (FiO_2_ 1 inspired at 1.4 ATA in a hyperbaric chamber) for high hyperbaric hyperoxia (very high hyperoxia, VHH). The authors observed, in the nucleus of the peripheral blood mononuclear cells (PBMC), that the return to normoxia after MH was sensed as a hypoxic trigger characterized by HIF-1α activation. By contrast, in the HH and VHH groups, there was a shift toward an oxidative stress response, characterized by nuclear factor E2-related factor 2 (NRF2) and nuclear factor-kappa B (NF-κB) activation in the first 24 h post-exposure. 

Interestingly, in another study evaluating the effects of hypoxemia on the human body, a similar increased activity of the transcriptional factor NRF2 (but not of NF-κB) was found after 24 h and 72 h of hypobaric hypoxia [21], highlighting again similar cellular responses of hyperoxia and hypoxia.

The same year, Khalife et al. [18] proposed a study similar to that by Fratantonio et al. [17] with apparently contrasting results. A group of 22 adult post-abdominal surgery patients was randomized to receive FiO_2_ 1.0 for one hour per day for eight consecutive days or no change in oxygen. Serum EPO, hemoglobin, and reticulocyte count were measured on admission to the ICU and on postoperative days seven and nine. EPO concentration at day nine was significantly higher in both groups compared to the baseline measurement on postoperative day 1. However, there were no differences between the groups in serum EPO concentration, hemoglobin, or reticulocyte count. As a possible confounder to these results, standard anesthesiology protocols were applied to both groups including the administration of FiO_2_ > 0.21 in the perioperative period, potentially influencing the results as seen in other works [16,17].

Debevec et al. investigated the effect of consecutive 1 h hyperoxic (FiO_2_ 1.0) and hypoxic (FiO_2_ 0.15) breathing interventions in 18 healthy adult individuals [55]. The authors found a reduction in EPO concentrations within the initial 8 h after the hyperoxic/hypoxic exposure compared to controls, suggesting that their results contradicted the normobaric oxygen paradox. However, the results should be interpreted in light of the findings of Fratantonio et al. [17], i.e., an excessive oxygen stimulus (such as FiO_2_ 1.0) does not elicit a paradoxical hyperoxic-hypoxic response during the same time-lapse. 

EPO elevation was found in scuba divers, performing a dive at a depth of 20–30 m (FiO_2_ about 0.6–0.8) for 30 min once a week for 5 consecutive weeks [59]. These results are coherent with the ones obtained by studying six scuba divers after a 14-day dive (8–10 m) breathing air at 1.8–2 ATA (roughly FiO_2_ 0.4) [57], and with the ones studying professional saturation divers, after decompression to the surface pressure, after long (from 25 to 27 days) saturation procedure at 80–90 m depth (Breathing Helium-Oxygen mixture (Heliox) FiO_2_ about 0.44) [58]. The recurrent exposure to normobaric oxygen breathing after hyperoxic conditions during diving could be the trigger for EPO production, even if other reasons may increase EPO production in this type of population which can even explain the lack in the increase of hemoglobin concentration (for example, the plasma volume changes during diving), as adequately pointed by the authors. 

In one of their most recent studies, Balestra et al. investigated the metabolic response to a single 1 h exposure to different FiO_2_ values (0.1, 0.15, 0.3, 1.0) at normobaric and hyperbaric conditions (1.4 ATA, 2.5 ATA), in 48 healthy subjects [20]. Blood samples were collected from each participant before and 120 min after oxygen exposure. The expression of microparticles (MPs) specific to platelets (CD41), neutrophils (CD66b), endothelial cells (CD146), and microglia (TMEM) was measured. There was a significant increase in MPs after all O_2_ exposures, except after mild hyperbaric (1.4 ATA) conditions, for which there was a significant decrease in MPs. Surprisingly, during the normobaric oxygen exposure, FiO_2_ 0.3 elicited similar responses to FiO_2_ 1.0.

## 6. Discussion

A hyperoxic stimulus followed by a return to a normoxic state seems to produce an organic response similar to that of a hypoxic steady state, prompting the expression of a variety of proteins, initiated by the activation of HIF-1a. The expression of HIF-1a represents an upstream response to the production of EPO, which is just one of the results of a real or mimic hypoxic state. Interestingly, this effect is faster than that needed to trigger EPO release during exposure to altitude and low atmospheric pressure (36–48 h vs. days, respectively) [60].

EPO has complex activities other than RBC production, including neuroprotection via its action as a neurotrophic factor in the central nervous system [61], cardioprotective properties [61], cardioprotective properties [62], and vasoactive effects through the increased production of endothelin [63], highlighting the potential beneficial effects of the hyperoxic-normoxic stimulus. Nevertheless, if it is true that a hyperoxic stimulus recreates the benefits of a hypoxic-like response without inducing a detrimental hypoxic status, it needs to be demonstrated that the benefit of this stimulus (short hyperoxic sessions) would overwhelm the known harmful effects of prolongued hyperoxia.

The studies presented in this review have yielded contrasting results regarding the use of this approach. First, EPO baseline is not easy to assess, and individual circadian rhythm may impact the results [53], so unique patient matching should be used in clinical trials. This approach was not followed in several studies, and the mean serum EPO values may mask real differences before and after a hyperoxic stimulus in the same subject. Second, the increase in EPO concentrations may be absent in some individuals due to the depletion of intracellular glutathione reserves; N-acetyl-l-cysteine (NAC) supplementation has been demonstrated to increase EPO with and without oxygen [64,65]. The explanation could lie in the fact that N-acetyl-l-cysteine (NAC) is a major precursor of glutathione [66], thus it is effective in promoting a redox balance within the cells [67]. In addition, it contributes to stabilizing HIF-1α subunit in the cytosol and thus favoring its translocation [68,69]. Overall, in case of depletion of the intracellular glutathione reserve NAC, recreating an adequate redox environment and stabilizing HIF results in EPO increasing [65]. Third, the optimum concentration and time of the oxygen stimulus needed to obtain the maximal response in EPO synthesis as well as the time needed between intermittent exposures to optimize the outcome are unknown. On the contrary, exposure to excessively high oxygen concentrations acts against the production of EPO [53] at least after similar post-exposure time lapses. Of course, if the outcome aims for a reduction of HIF stimulation, higher concentration, and probably closer repetitions, may be needed. Finally, in some clinical studies, EPO serum concentration had similar changes in both the oxygen group and the control group (not exposed to high levels of extra oxygen). It does not necessarily go against the normobaric oxygen paradox since even minor O_2_ variations (e.g., routine post-operative oxygen supply) may trigger the mechanism

An intermittent normobaric hyperoxic stimulus elicits a hypoxic-like response as it has been proven in the study of Cimino et al. [14]. The increase in the activity of HIF represents an upstream response to the production of EPO and Hemoglobin, which is just one of the results of a hypoxic real or mimic state. Anyway, if it is true that a hyperoxic stimulus re-creates the benefits of a hypoxic-like response without inducing a detrimental hypoxic status, it still has yet to be proven that the benefit of this stimulus overwhelms the known harmful effects of hyperoxia although such a short time of exposure is not likely to enter toxic levels, especially in pathologic situations.

### Future Perspectives

Although the strength of this stimulus must still be measured, this paradox could have several clinical implications, from the treatment of anemia, thus limiting blood transfusions [70], to adjuvant therapy for septic patients [71], to precondition agent for sports training [72,73], to therapy in critical care settings in which a HIF response without an actual hypoxic state could be effective in cardio and neuroprotection [74,75].

Another potential future field of application concerns the therapeutic role of oxygen in tumor growth and spread. New evidence about oncologic progression has shown that cancer cells may evolve into a hypermetabolic state, which can be sustained even in the presence of a limited oxygen supply [76,77]. Genetical changes and uncontrolled cancer growth, which generate intra-tumor hypoxic areas (used by the innovative hypoxia-responsive drug delivery nanoplatforms [78]), cause HIF-1α overexpression enhancing neo-angiogenesis through VEGF synthesis and therefore favoring cancer progression and metastasis [79]. Only recently has the possible role of hyperoxia in tumor necrosis or development started to be investigated [80,81]. For example, leukemia cell lines exposed to a hyperoxic stimulus have been shown to increase the expression of caspase 3, and committing themselves to programmed death and apoptosis preceded morphological modifications of T and B cells [82]. 

In an animal model of breast cancer, Raa et al. [83] observed that normobaric and hyperbaric oxygen treatments showed more efficacy than conventional chemotherapy alone in reducing tumor growth, limiting neo-angiogenesis, decreasing cancer vascularization and inducing apoptosis. 

The different response to the increased oxygen concentration in cancer cells, compared to healthy cells, may be linked to several mechanisms: A maladaptive response to hyperoxia, as well as to modification in adenosine pathways able to trigger anticancer effects of T cells and NK cells [84,85];Incapacity to deal with the overproduction of ROS during hyperoxia, ultimately leading to necrosis and apoptosis [83];A disequilibrium between antioxidants and reactive oxygen species [86].

## 7. Conclusions

Oxygen is a drug that is used in patients around the world on a daily basis. It is inexpensive and generally safe and may have more beneficial effects than just increasing the blood oxygen content. In the future, oxygen therapy could be implemented as a coadjutant in new frontiers of therapeutic schemes to treat several diseases. Evidence shows that a dynamic change in oxygen caused by a normobaric hyperoxia stimulus has a different effect to that of a steady state. The optimal timing and intensity of the O_2_ stimulus still have to be determined. Efforts should be directed to further confirm this phenomenon, as a promising, cheap, and easy-access contribution in several clinical situations.

## Figures and Tables

**Figure 1 ijms-24-00082-f001:**
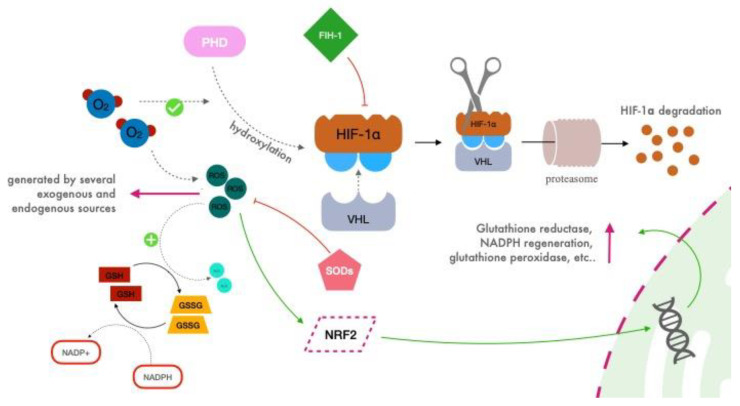
Simplified schematic representation of HIF activity during normoxia.

**Figure 2 ijms-24-00082-f002:**
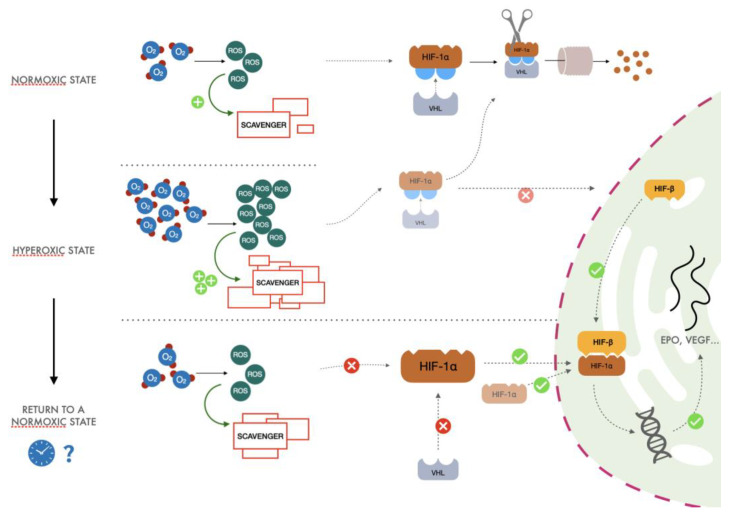
A simplified description of the proposed normobaric oxygen paradox.

**Figure 3 ijms-24-00082-f003:**
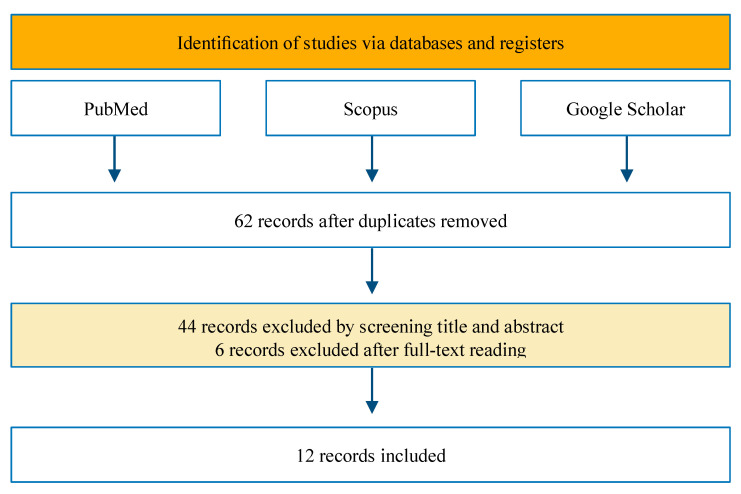
Flow diagram of the review.

**Table 1 ijms-24-00082-t001:** A summary of the main human studies on the normobaric oxygen paradox.

Author (Year)	Type of Study	No. of Patients	Intervention	Main Results
Balestra (2006) [15]	Human experimental study	16 healthy adults	Exposure to normobaric oxygen at FiO_2_ 1.0 for 2 h vs. exposure to hyperbaric oxygen 2.5 ATA FiO_2_ 1.0 for 1.5 h	Increase in EPO after normobaric oxygen exposure and decrease in EPO after hyperbaric oxygen exposure
Keramidas (2011) [53]	Single-blinded experimental study	10 healthy males	Exposure to normobaric oxygen at FiO_2_ 1 for 2 h × 7 d	Decrease in EPO levels after hyperoxic exposure compared with control group
Ciccarella (2011) [54]	Double-blind prospective pilot study	20 post cardiac surgery patients who had intraoperative CPB and MV	Exposure to normobaric oxygen at FiO_2_ 1.0 for 2 h vs. FiO_2_ 0.5 for 2 h	Increase in EPO in both groups but slope of the increase in the EPO plasma level significantly higher in those exposed to hyperoxia and relative hypoxia
Debevec (2011) [55]	Human experimental study	18 healthy male adults	Single exposure to 1 h normobaric oxygen FiO_2_ 1.0 followed by 1 h normobaric FiO_2_ 0.15	Exposure to hyperoxia followed by mild hypoxia led to temporary decrease in EPO levels. No difference in late time points for EPO levels Compared to control group
Lafere (2013) [56]	Double-blind multicenter clinical study	85 ASA 1 and 2 patients undergoing surgery for traumatic hip fracture	Exposure to 30 min of FiO_2_ 1.0 normobaric oxygen vs. Air from POD 1 until discharge	Increase in reticulocytes count and reduction in hospital LOS and RBC transfusion in the experimental group.
Revelli (2013) [57]	Human experimental study	6 scuba divers	14-days of diving (8–10 m) with air at 1.8–2 ATA	Significant rise in serum EPO observed at 24 h post emersion
Donati (2017) [19]	Prospective observational pilot study	40 mechanical ventilated patients	Exposure to normobaric oxygen at FiO_2_ 1.0 for 2 h	ROS increase after 1 h and glutathione level after 2 h from hyperoxia exposure. Reduction of microvascular density and perfusion during oxygen exposure rapidly normalized after returning to ambient air. EPO level rise after 48 h.
Kiboub (2018) [58]	Human experimental study	13 scuba divers	Decompression to surface pressure after long (from 25 to 27 days) professional saturation dive at 80–90 m depth	EPO markedly increased within 24 h after decompression
Perović (2020) [59]	Human experimental study	14 scuba divers	One dive per week over 5 weeks at a depth of 20–30 m for 30 min	A significant EPO increase before and after the third and the fifth dive compared to the level before and after the first dive.
Fratantonio (2021) [17]	Human experimental study	12 healthy adults	1 h exposure to normobaric oxygen FiO_2_ 0.3 vs. normobaric oxygen FiO_2_ 1.0 vs. hyperbaric oxygen 1.4 bar FiO_2_ 1.4	Exposure to lower level of FiO_2_ associated with a stronger response in HIF1-α synthesis and a lower level of inflammation and oxidative stress which was also less persistent than exposure to hyperbaric 1.4 FiO_2_ oxygen
Khalife (2021) [18]	Prospective randomized clinical study	26 female patients undergoing breast surgery	1 h per day of normobaric oxygen FiO_2_ 1.0 from POD 1 for 8 consecutive days	No difference in EPO or hemoglobin levels between the groups
Balestra (2022) [20]	Human experimental study	48 healthy adults	Single 1 h exposure to FiO_2_ 0.10, 0.15, 0.3 or 1.0 normobaric oxygen and 1.4, 2.5 ATA hyperbaric oxygen	Significant elevation in microparticles from different cells was observed after exposure to every different oxygen concentration except after hyperbaric 1.4 ATA oxygen exposure.

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
