# Peer review of "The Normobaric Oxygen Paradox—Hyperoxic Hypoxic Paradox: A Novel Expedient Strategy in Hematopoiesis Clinical Issues"

_ijms, 2022, doi:10.3390/ijms24010082_

Round 1
Reviewer 1 Report
The article “The Normobaric Oxygen Paradox – Hyperoxic hypoxic paradox: a novel expedient strategy in hematopoiesis and vascular clinical issues” overviewed the physiological processes behind the proposed normobaric oxygen paradox, focusing on the latest scientific progress and its potential clinical applications. In recent years, intermittent switching between hyperoxic and normoxic exposure has been proved to boost the production of erythropoietin and increase hemoglobin levels as a new preconditioning strategy for frail patients and prevention of postoperative anemia. In this manuscript, the potential application of normobaric Oxygen Paradox – hyperoxic hypoxic paradox in other fields such as tumor is summarized and discussed. In general, this manuscript is well organized and written and I would recommend the publication of the manuscript after the following issues are addressed:
1. On page six, the font of NF-kB is different from the rest of the font that uses the proper noun throughout the article.
2. On page 11, the second paragraph is not marked with quotations, so it needs to add quotation notes (“An intermittent normobaric hyperoxic stimulus elicits a hypoxic-like response as it has been proven in the study of Cimino et al”).
3. It is mentioned in the introduction that this paper pays special attention to the clinical problems of hematopoietic and vascular physiology. However, the introduction of vascular physiology in this manuscript seems to be too concise, which can be appropriately supplemented.
4. Please add some description upon the arrows in the figures to make it more informative. For example, how is the oxygen converted into ROS in Figure 1? What does the arrow between PHD and HIF-1α represent (hydroxylase I guess?)?
5. Please cite some relevant works in the manuscript “Kuikun Yang,# Guocan Yu,# Rui Tian, Zijian Zhou, Hongzhang Deng*, Ling Li, Zhen Yang, Guofeng Zhang, Dahai Liu, Jianwen Wei, Ludan Yue, Ruibing Wang* and Xiaoyuan Chen*, Oxygen-Evolving Manganese Ferrite Nanovesicles for Hypoxia-Responsive Drug Delivery and Enhanced Cancer Chemoimmunotherapy. Adv. Funct. Mater. 2021, 31, 2008078.”
Author Response
The article “The Normobaric Oxygen Paradox – Hyperoxic hypoxic paradox: a novel expedient strategy in hematopoiesis and vascular clinical issues” overviewed the physiological processes behind the proposed normobaric oxygen paradox, focusing on the latest scientific progress and its potential clinical applications. In recent years, intermittent switching between hyperoxic and normoxic exposure has been proved to boost the production of erythropoietin and increase hemoglobin levels as a new preconditioning strategy for frail patients and prevention of postoperative anemia. In this manuscript, the potential application of normobaric Oxygen Paradox – hyperoxic hypoxic paradox in other fields such as tumor is summarized and discussed. In general, this manuscript is well organized and written and I would recommend the publication of the manuscript after the following issues are addressed.
R0) We thank the Reviewer for the positive consideration of our work.
- On page six, the font of NF-kB is different from the rest of the font that uses the proper noun throughout the article.
R1) We apologize for the mistake and thank the reviewer for allowing us to correct ourselves. The font has been corrected as recommended.
- On page 11, the second paragraph is not marked with quotations, so it needs to add quotation notes (“An intermittent normobaric hyperoxic stimulus elicits a hypoxic-like response as it has been proven in the study of Cimino et al”).
R2) According to the Reviewer’s comment, the quotation note has been added.
- It is mentioned in the introduction that this paper pays special attention to the clinical problems of hematopoietic and vascular physiology. However, the introduction of vascular physiology in this manuscript seems to be too concise, which can be appropriately supplemented.
R3) We thank the Reviewer to rise this well perceived remark; after a discussion with the other authors, we think that it could be better to remove the word “vascular” from the title and the manuscript, as the evidence from the studies reported is not really investigated in that field and globally more focused on the hematopoietic area.
- Please add some description upon the arrows in the figures to make it more informative. For example, how is the oxygen converted into ROS in Figure 1? What does the arrow between PHD and HIF-1α represent (hydroxylase I guess?)?
R4) Figure 1 has been redesigned following the Reviewer's suggestions to make it more informative. Thank you for this!
- Please cite some relevant works in the manuscript “Kuikun Yang,# Guocan Yu,# Rui Tian, Zijian Zhou, Hongzhang Deng*, Ling Li, Zhen Yang, Guofeng Zhang, Dahai Liu, Jianwen Wei, Ludan Yue, Ruibing Wang* and Xiaoyuan Chen*, Oxygen-Evolving Manganese Ferrite Nanovesicles for Hypoxia-Responsive Drug Delivery and Enhanced Cancer Chemoimmunotherapy. Adv. Funct. Mater. 2021, 31, 2008078.
R5) According to the Reviewer’s suggestion, this relevant and interesting work has been cited in the manuscript.
Reviewer 2 Report
With the available literature , it is generally believed that a higher hypoxic stimulus would trigger HIF-1α regulated gene synthesis cascade with higher EPO and Hb levels. Paradoxically, this is not the only mechanism for triggering such cascade, hence this term “normobaric oxygen paradox-NOP” has been used for the first time by the authors. This needs further studies for revalidation on a large population. However, such future studies has nothing to do with publication of this review article.
The review meticulously summarizes important human studies on NOP with all the possible molecular and cellular mechanisms involved and describes the activation of HIFs by NOP administration that stimulates even faster than altitude exposure and low atmospheric pressure, thus opens new vistas in environmental physiology and clinical medicine. Authors may think to undertake comparative studies in future in this direction.
This review highlights absence of increase of EPO in some individuals due to the depletion of the intracellular glutathione reserve. The same may be described in details with citation of few more references.
Author Response
With the available literature, it is generally believed that a higher hypoxic stimulus would trigger HIF-1α regulated gene synthesis cascade with higher EPO and Hb levels. Paradoxically, this is not the only mechanism for triggering such cascade, hence this term “normobaric oxygen paradox-NOP” has been used for the first time by the authors. This needs further studies for revalidation on a large population. However, such future studies has nothing to do with publication of this review article. The review meticulously summarizes important human studies on NOP with all the possible molecular and cellular mechanisms involved and describes the activation of HIFs by NOP administration that stimulates even faster than altitude exposure and low atmospheric pressure, thus opens new vistas in environmental physiology and clinical medicine.
R7) We thank the Reviewer for the positive consideration of our work.
Authors may think to undertake comparative studies in future in this direction.
R8) Yes, we would like to further investigate this phenomenon and comparative studies are of great interest and already planned in the next future.
This review highlights absence of increase of EPO in some individuals due to the depletion of the intracellular glutathione reserve. The same may be described in detail with citation of few more references.
R9) We thank the Reviewer for raising this important point. Therefore, a more explicit referenced description has been added accordingly.